# Child Maltreatment and Intimate Partner Violence in Mental Health Settings

**DOI:** 10.3390/ijerph192315672

**Published:** 2022-11-25

**Authors:** Jill R. McTavish, Prabha S. Chandra, Donna E. Stewart, Helen Herrman, Harriet L. MacMillan

**Affiliations:** 1Offord Centre for Child Studies, Department of Psychiatry and Behavioural Neurosciences, McMaster University, 293 Wellington St. North, Hamilton, ON L8L 8E7, Canada; 2NIMHANS Hospital, Hosur Rd, near Bangalore Milk Dairy, Hombegowda Nagar, Bengaluru 560029, Karnataka, India; 3Centre for Mental Health, University Health Network, 200 Elizabeth St, 7EN229, Toronto, ON M5G 2C4, Canada; 4Centre for Youth Mental Health, The University of Melbourne, Parkville, VIC 3052, Australia; 5Orygen, Parkville, VIC 3052, Australia; 6Department of Pediatrics, McMaster University, Health Sciences Centre 3A, 1280 Main Street West, Hamilton, ON L8S 4K1, Canada

**Keywords:** child maltreatment, intimate partner violence, mental health, training, clinical responses

## Abstract

Intimate partner violence (IPV) and child maltreatment (physical, emotional, sexual abuse, neglect, and children’s exposure to IPV) are two of the most common types of family violence; they are associated with a broad range of health consequences. We summarize evidence addressing the need for safe and culturally-informed clinical responses to child maltreatment and IPV, focusing on mental health settings. This considers clinical features of child maltreatment and IPV; applications of rights-based and trauma- and violence-informed care; how to ask about potential experiences of violence; safe responses to disclosures; assessment and interventions that include referral networks and resources developed in partnership with multidisciplinary and community actors; and the need for policy and practice frameworks, appropriate training and continuing professional development provisions and resources for mental health providers. Principles for a common approach to recognizing and safely responding to child maltreatment and IPV are discussed, recognizing the needs in well-resourced and scarce resource settings, and for marginalized groups in any setting.

## 1. Introduction

Intimate partner violence (IPV) and child maltreatment are two of the most common types of family violence; they are associated with a broad range of consequences, including morbidity and mortality. IPV is defined as any physical, psychological, or sexual harm committed by a current or former partner or spouse [1]. Child maltreatment can include physical, emotional, or sexual abuse, as well as neglect and children’s exposure to IPV [2]. Global estimates suggest that one in three women (30%) have experienced either physical and/or sexual IPV in their lifetime, one in four adults (23%) were physically abused as children, one in nine adults (12%) were sexually abused as children, and one in six older adults (15.7%) have been abused [3,4,5]. These prevalence rates are even higher in clinical settings [6,7,8,9], suggesting many mental health clinicians work with individuals who are exposed to family violence. In this paper we focus on mental health settings and discuss the needs in these settings for awareness of the high prevalence of child maltreatment and IPV, and for safe, effective and co-designed frameworks for recognizing and responding to people presenting with these experiences. While there are important overlaps between child maltreatment and IPV and other forms of violence, such as elder abuse [10], teen dating violence [11], cyber dating violence [12], and child sex trafficking [13], this paper does not specifically address these other forms of violence.

Experiences of one type of family violence increase the risk of experiencing another type, as children experiencing maltreatment are more likely to experience other forms of violence, including IPV [9,14,15]. Furthermore, the health consequences of family violence are significant. For example, child maltreatment has the potential to negatively impact individuals across the lifespan, with developmental delay first seen in infancy; anxiety and mood disorder symptoms and poor peer relationships first seen in childhood; substance use to the point that a person has problems occurring, develops an addiction or is diagnosed with a substance use disorder often first seen in adolescence; and increased risk for personality and other psychiatric disorders, relationship problems, and maltreatment of one’s partner or children in adulthood [7,16,17,18,19,20].

Gender-based violence refers to “harmful acts directed at an individual based on their gender” and can include forms of violence like IPV and child maltreatment, as well as other forms of violence, such as child marriage and honour crimes [21]. Gender-based violence acknowledges that people experience family violence at different rates depending on several social determinants of health, including gender. For example, studies have shown that women experiencing IPV are far more likely to have higher rates of serious injury and mortality [5]. Studies have also shown that rates of IPV are “dramatically” higher for transgender individuals [22]. Rates of IPV in other LGBQ+ groups are also found to be equal or greater than in heterosexual couples, potentially due to minority stress or internalized homophobia, which is associated with increased risk for both violence victimization and perpetration [23,24]. Gender-based differences are also seen with respect to child maltreatment, as girls are more likely to experience sexual abuse and boys are more likely to experience physical abuse [25,26].

## 2. Clinical Features of Child Maltreatment and IPV

While family violence is not always obvious, recognizing it in clinical encounters is critical. Any time providers are concerned about a patient’s social, emotional, or behavioural responses, they should consider what environmental factors, including family violence, could be influencing these responses.

For child maltreatment, recognition involves being aware of the broad range of signs and symptoms of potential maltreatment, as well as potential risk factors. Some research suggests providers have an easier time recognizing physical signs of maltreatment, such as bruises or broken bones [27]. The National Institute for Health and Care Excellence (NICE) has delineated over 70 indicators of child maltreatment [28]. However, some children who are experiencing maltreatment show no signs or symptoms of maltreatment; there may only be risk factors, such as IPV in the home. Risk factors for child maltreatment can cross socioecological levels and include child risk factors (e.g., presence of disability), parental risk factors (e.g., parental alcohol use or mental ill-health), relationship risk factors (e.g., poor attachment between parent and child, poor family functioning), or community risk factors (e.g., high poverty). Some children who are experiencing maltreatment have no risk factors for maltreatment; they may only have non-specific symptoms, such as depression. Finally, for some children, providers may only see a concerning interaction between a parent and child, such as a parent screaming at an infant.

Consultation may be needed when providers are considering potential maltreatment, due to the complexity of its presentation. For example, when assessing for and treating potential neurodevelopmental disorders, such as attention deficit hyperactivity disorder (ADHD), clinicians should be alert to symptoms that overlap with child maltreatment (e.g., agitation, poor self-esteem, difficulties concentrating, and difficulties with work, school and sleep) [29,30,31,32,33]. Experts have noted that pharmacological treatment for ADHD when the symptoms are resultant from maltreatment does not allow for treatment of relational injuries that are underlying symptoms [31], but also that a potential ADHD diagnosis may need to be revisited after effective treatment for traumatic stress [32]. It is also important for mental health providers to recognize that globally, persons with disabilities, including children and teens, have increased risk of all forms of violence [34]; for example, children and adults with disabilities are twice as likely to experience sexual violence in their lifetime [35]. While experiences of physical and sexual violence against people with disabilities is a major risk factor for ill health, recognition of and safe response to these experiences remains low [36]. These are examples of the complexity of recognizing child maltreatment.

Similarly, IPV is not always obvious, and people may present to providers with non-specific indicators, such as depression or chronic pain. Important indicators of potential IPV include behavioural signs (e.g., repeated cancelled visits, partner insists on being present for all aspects of the assessment and resists leaving); the person’s physical health problems (e.g., injuries, chronic pain, functional disorders), reproductive health problems (e.g., gynecological disorders, sexual dysfunction, unwanted pregnancies), or mental health problems (e.g., depression, anxiety, post-traumatic stress symptoms); and risk factors for the person experiencing violence and their partner (e.g., past history of family violence), the partner (e.g., depression or substance use), the relationship (e.g., recent separation), or the community (e.g., social isolation). For IPV it can be important to distinguish bilateral violence, which is considered a less severe form of violence, from battering or intimate partner terrorism [37], where the “partner uses violence and other coercive control tactics to attempt to take general control over his or her partner” (p. 187). As coercive control is associated with increased risk of suicidality, anxiety and depression, and higher rates of physical and sexual violence for the person experiencing violence [9,38,39], mental health providers need to consider the interrelationship between severe forms of violence and significant mental health symptoms. For both child maltreatment and IPV, the more indicators are present, the more likely family violence is a concern.

## 3. Trauma- and Violence-Informed and Rights-Based Care

Human rights violations in mental health care have been identified as a global emergency [40]. A human rights-based approach to mental health care [40] “means placing emphasis not only on avoiding human rights violations but making sure that human rights principles are at the center of a service-providing organization” (p. 264). It is increasingly recognized that people with lived experience of violence must be involved with, consulted about, or co-producers of interventions, so that services and community-based supports best match their needs and concerns [9,41,42,43]. Furthermore, while the rights of adults are increasingly recognized, it is important to consider children’s rights in all aspects of service delivery. This can include, for example, their right to know (e.g., why they were placed in care, if relevant), right to be prepared (e.g., for changes to services), and right to be involved in accordance with their preferences, age, and developmental stage [44].

In addition to a rights-based approach to mental health care, trauma-informed care is an increasingly familiar approach used in mental health services to acknowledge that any patient presenting for care may have a history of trauma [45,46,47]. Given the high prevalence and serious negative impacts of trauma, principles of trauma-informed practice are increasingly incorporated into core competencies for mental health providers [48,49,50]. These principles address the high prevalence of trauma in those presenting to mental health settings, the need to tailor assessments and interventions to the person’s experience of trauma, and the recognition of how trauma can impact person-provider interactions. For example, children and adults with experiences of trauma may have difficulty trusting others, including healthcare providers. Providers need to demonstrate behaviours that enhance people’s felt sense of safety and trust by, for example, “providing a therapeutic alliance that fosters trust and interpersonal security” [48].

Trauma and violence-informed care (TVIC) “extends the trauma-informed care framework with the addition of ‘v’iolence to emphasize the association between trauma and violence” [51]. TVIC acknowledges the high prevalence of violence experiences, particularly in mental health settings, as well as the interlocking and additive effects of violence (e.g., experiences of interpersonal violence, such as child maltreatment, can increase risk of other forms of interpersonal violence; overlap with systemic and structural violence, such as racism, casteism, and colonialism; and the interlocking and additive effect of these experiences of violence, which increase risk for negative outcomes) [51,52]. Systemic violence has only recently started to be taken up by healthcare researchers and has shown to be associated with healthcare accessibility and negative health outcomes [53,54,55,56]. The © Violence, Evidence, Guidance, Action (VEGA) Environment, Approach, Responses (EAR) model provides an accessible way to learn and practice TVIC, as it addresses the importance of listening to patients and attending to safety in the environment, the provider’s approach to care and specific provider responses to patients [51]. For example, safety in the environment might entail a private space to have conversations about suspected experiences of family violence; safety in the approach acknowledges such factors as the importance of informed consent, especially regarding the limits of confidentiality; and safety in the provider’s approach might involve specific questions about family violence that are asked in a way to increase patient’s safety (discussed further below). The TVIC approach also acknowledges the role of the healthcare system in supporting individual providers. For example, in a recent meta-synthesis [57] of 46 articles assessing healthcare providers’ readiness to address IPV, the authors found that the majority of articles indicated providers needed support from the healthcare system. Common mechanisms of healthcare system support asked for by providers included (a) support to upskill (i.e., training) regarding identification and response to IPV; (b) support to include questions about IPV in the assessment process, and (c) policies and procedures to support the identification and response to IPV in everyday practice [57]. Enhancing healthcare system responses to people’s experiences of child maltreatment and IPV is essential to a TVIC approach to care [9,57].

While TVIC frames the entire care encounter—from before the patient encounter (e.g., is the provider trained in safe responses to family violence?), the moment the patient walks into the building (e.g., is the environment welcoming and safe?) to the care encounter (e.g., does the provider engage with the client in a manner that enhances safety?)—this article will focus primarily on the care encounter.

## 4. Asking about Child Maltreatment or IPV

For both child maltreatment and IPV, there is no evidence that screening (asking every patient about potential experiences of family violence in a standardized way regardless of their presentation) improves the patient’s outcomes or reduces their experience of family violence, so screening is not recommended [58,59]. It is increasingly acknowledged how standardized screening or referral procedures that are intended to be helpful (e.g., automatic referral to child welfare if a previous child has been involved with child welfare) can compound intergenerational trauma, leading to increased child removals, increased child welfare involvement, increased avoidance of preventive services and with no evidence of associated benefit to children and families [60]. Instead of screening, in the context of a mental health assessment, the provider will ideally include an appropriately phrased enquiry about family violence, when safe to do so, as part of the regular history-taking in a clinical encounter.

The provider should ask the child/adult about potential Ies of family violence in a way that is tailored to their presentation. Before doing so, the provider should take a number of steps in order to enhance the child/adult’s safety, including (1) taking training on how to ask and respond safely to family violence; (2) securing a private space in the setting; (3) developing an approach to making referrals within or outside of the organization, should family violence or other environmental factors be present; and (4) if needed, using professional interpreters and not family or friends to speak with the patient [51]. In many situations, providers recognize the adult’s need for privacy but the child’s need for privacy is not acknowledged. As the child may be accompanied by a caregiver who has harmed or put the child at risk, children should not be asked about maltreatment in front of their caregivers; the provider should speak to the child on their own when asking about child maltreatment [51].

Safe inquiries about potential family violence experience must involve discussions about the limits of confidentiality. In potential cases of maltreatment, it is important to discuss the limitations of confidentiality with children in a developmentally appropriate way [51]. For example, providers might say something like “what we talk about today is private unless I’m concerned about safety. What is your understanding of safety?” [51]. By asking children about their understanding of safety, providers have an opportunity to clarify what types of safety concerns might warrant breaking confidentiality. For example, in response to the child, providers might say “For me safety involves someone being hurt or not taken care of. If you, or someone you know, is hurting themselves or being hurt by someone else then that’s a problem with safety. If someone is not being looked after, that’s also a problem with safety. In these cases, I may need to contact someone whose job it is to help keep people safe” [51]. It is important when considering potential experiences of maltreatment to be aware of cultural differences in parenting. For example, in some Indigenous communities, the wider family network or community share the responsibility to keep children safe along with the individual biological parents [60] and as such assessments of a child’s care and wellbeing should consider the wider family network. It is also desirable to tap into community wisdom and counsel, especially in communities exposed to historical violence and losses [60].

Following discussions of limits of confidentiality, a phased inquiry approach [61] to questions about family violence can be undertaken. Research has shown that children and adults prefer a phased inquiry approach [61], or an approach that first addresses the patient’s presenting concern (e.g., cough, bruise, anxiety), then addresses the patient’s general wellbeing (e.g., how are things at school/work?), then addresses safety in the home (e.g., how is everyone getting along at home?). More specific questions can be asked depending on the patient’s responses to the general questions as well as their presentation; these questions should be tailored to their age, developmental stage, specific presentation [51], and any cultural expectations about parenting [62]. For example, for concerns about child physical abuse a provider might ask “What happens at home when people get mad or angry?” [51]. For concerns about IPV, a provider might ask: “How are things at home?”, “How do you and your partner get along?”, [51] or “Is there anyone at home that uses drugs or alcohol?” and (if yes) “What happens when they are intoxicated?”.

Questions about potential experiences of child maltreatment or IPV should be culturally sensitive and attuned to the context of the person. For example, in some families violence may be committed or condoned by extended family members [63,64,65] and there may be many cultural barriers to seeking help [66], such as “limited awareness of rights, emphasis on the notion of ‘honor’, negative societal attitudes towards women, [and] the economic dependence of women on their husbands and families” (p. 23). Furthermore, in some countries there are no laws to forbid certain experiences of maltreatment or IPV; instead, support against violence comes from families [66]. In these situations, positive and negative roles of family, including extended family, must be considered and assessed.

## 5. Responding Safely to Child Maltreatment or IPV

Safe responses to child maltreatment build upon providers’ communication skills, such as the use of active listening, compassion, and developmentally appropriate language. Many aspects of the LIVES (listen, inquire about needs and concerns, validate, enhance safety, and support) approach to safe responses to IPV (discussed below) can be applied to children experiencing maltreatment with the exception of validation of maltreatment. As in many jurisdictions it is the responsibility of child protection services to determine if maltreatment has occurred (further discussed below); it is important for providers to recognize the limitations of their role in this regard [51]. However, empathetic provider responses can validate that no one should experience maltreatment without “confirming” the experience as maltreatment, such as “I’m sorry that happened to you, it is not okay that you were hit” [51]. Disclosures of maltreatment will lead to mandatory reporting to child protection services, where mandatory reporting is in place, which should be communicated to the child in a sensitive way. For example, “If you remember we spoke about how I might need to contact someone if there was a concern about someone’s safety. What you have shared with me is an example of a concern about safety” [51].

A meta-synthesis has indicated that women experiencing IPV want providers to be nonjudgmental, nondirective and to tailor their responses to the individual [67]. Women expressed that they wanted the care interaction to “progress at their own pace and not to be pressured to disclose, leave the relationship, or press charges against their partner or ex-partner” [67]. The LIVES approach was developed by the World Health Organization (WHO) as a way to organize these and other qualities of a safe, initial response to disclosures of IPV [68]. No matter what the provider’s role is or the time constraints of the encounter, a safe initial response using the LIVES approach is essential. The five tasks of the LIVES approach included the following elements. Listening involves active, compassionate listening skills which are at the foundation of many providers’ practice, including but not limited to concentrating while listening, using nonverbal communication (e.g., facial expressions, head nods), not interrupting the speaker, asking questions (including using open-ended questions), and understanding the speaker’s feelings and emotions [69]. Inquiring about needs and concerns requires assessing and responding to the persons’ physical, social, and practical needs and concerns. Validation involves the provider demonstrating that they are taking the information that the person has shared seriously and involves discussions regarding how everyone deserves to be safe. Enhancing safety focuses on discussions with the patient about how to protect them from future harm (discussed further below). Support refers to discussing the patient’s needs and priorities, the ways they have kept themselves safe up to this point, and help to connect them to information, services, and support.

### 5.1. Assessing Immediate Safety

A safe initial response to IPV involves an assessment of the patient’s immediate safety and, if children are involved, their safety as well [51]. It is critical to assess whether it is safe for the person being abused to return home. Some ways that you can ask about safety in the home include, “Are you concerned about your safety or the safety of children in the home?” or “Have there been any significant changes in your (ex)partner’s life, such as a job loss or increased substance use?” [51]. If you or the patient you are speaking with are concerned about their immediate safety, it is important to tell the patient about the risk factors you are aware of that suggest increased risk of harm and to discuss your concerns about their safety. While it is not possible to predict if IPV will become fatal, there are a number of warning signs that, if present, require discussion with the patient. A recent meta-analysis has summarized some of these risk factors for women’s fatality in relation to abuse perpetrated by men, such as: if the abusive (ex)partner has direct access to guns or “had previously threatened the victim with a weapon, had previously strangled the victim, had threatened to harm the victim, had perpetrated forced sex, exhibited controlling behaviors…abused the victim while she was pregnant, previously stalked the victim, was jealous, abused substances, had less than a high school education, was younger in age, had anger problems, and had a history of mental health issues” [70]. While these meta-analysis findings are specific to heterosexual couples, many of these risk factors (e.g., access to guns or other weapons, previous strangulation, forced sex, stalking, use of substances) would also be clinically relevant to discuss (if present) with patients who are experiencing IPV in LGBTQ+ relationships.

You can also discuss options with the person if it is not safe to return home, such as contacting a local IPV shelter for further consultation or another referral source that may be useful to the patient. Except for contacting child protection services (when there are children in the home), when necessary, it is important to obtain the patient’s consent to contact services, such as the shelter or the police. If the person does not consent to you contacting another service but you are concerned about their immediate safety, document your concern but also respect their wishes. Doing otherwise would breach confidentiality [51].

### 5.2. Evidence-Based Interventions

A safe response to family violence is likely to involve a multidisciplinary, evidence-informed, clinical and community-based action that includes referrals and resources provided by several partners. Supporting a family to prevent future experiences of violence is a separate but important role from supporting a family to address any health consequences of family violence. For children, prevention of future experiences of violence should be done in coordination with child welfare services, where these services are in place. It is important to note that the assessment of violence, including IPV between caregivers, is not solely the responsibility of child welfare [32]. Providers, including mental health providers, also share a role in safety assessments.

For children, whenever possible, taking safety into account, referral to evidence-based services should take place after a comprehensive assessment has identified appropriate referral(s) that are tailored to the unique needs of the child, including interventions that are designed for their age and developmental stage [32]. A comprehensive assessment addresses meaningful aspects of the child’s life, such as their home situation (e.g., description of people in the family, present living situation, extended kinship networks, any non-traditional familial relationships); their education (e.g., school, grades, teachers); their involvement with activities (e.g., recreation); their mental health; and any other relevant aspects of their lives (e.g., substance use, sexual health) [51]. For assessments with children, it is important to observe them with their caregiver and to also communicate with them separately from their caregiver [51]. It is also important to gather information about other people in the child’s life, such as siblings, as well as information about caregivers (e.g., their personal, social and health history; their family history, including their experiences of being parented; any adverse childhood experiences; and the quality of their relationship with the child) [51]. Gathering this information is necessary for accurate case formulation [71]. It also helps to shift the emphasis from “what is wrong with you” to “what has happened to you,” a trauma-informed strategy intended to reduce stigma and validate the experiences of the child and family [32]. In situations (e.g., counseling) where initial assessments are done with parents alone or with parents and children, clinicians should consider how caregivers’ own mental health concerns and conditions may impact ongoing treatment with children. If a caregiver is not able to be an emotionally safe and regulated person in the room with the child, child-only therapy may be indicated or, for young children (e.g., under the age of 5), referral to an evidence-based attachment intervention to increase the caregiver’s safe attachment behaviours may be indicated [72,73]. Referrals for the caregiver’s own support (e.g., primary physician, their own counseling) may also be necessary to support the well-being of the child. For example, treatment options for a child who has been physically abused by a parent may differ depending on if the parent acknowledges or denies that they abused the child, if the parent has co-occurring mental health concerns, if the child has a disability or other complex health need, if the child has internalizing or externalizing symptoms, how far from treatment the family lives, the present living situation (e.g., if homelessness is a pressing concern, housing stability may be a priority), and so on.

Some evidence-based treatments for children who have experienced maltreatment do exist. For example, children who have experienced sexual abuse or exposure to IPV and who have post-traumatic stress symptoms may benefit from cognitive behavioural therapy with a trauma focus [74,75,76]. Children who have experienced physical abuse or neglect and who have externalizing problems may benefit from parent–child interaction therapy [77,78]. Other evidence-based interventions should be considered if the assessment of the child identifies other types of mental health symptoms or disorders (e.g., depression). Support for the family so that safety, love, and nurturing can be prioritized for the child should include considerations of (a) access to culturally responsive, trauma-informed services, (b) partnerships within over-represented child welfare communities to develop culturally relevant care, and (c) active participation by children and families in decision making [60]. It is important for providers to identify culturally and language-appropriate supports and programs for children that build on their strengths, talents and skills [79], including recreational activities and school-related programs. Following a report to child welfare, it is important to follow up with children and adults experiencing family violence to see if referrals have been useful, if experiences of family violence have stopped, or if other services and supports are needed.

Most research about supports for people who have experienced IPV focus on women who have been abused by men; additional research is needed regarding appropriate evidence-based supports for people in LGBTQ+ relationships who are experiencing IPV. Women who have experienced IPV may benefit from being referred to brief, structured advocacy interventions [80,81,82]. These interventions may reduce physical abuse and may provide small short-term mental health benefits, especially to pregnant women or women experiencing less severe abuse [80]. Various psychological therapies, such as cognitive behavioural therapy [82,83,84], may be supportive to patients who are experiencing specific mental health symptoms, such as depression or anxiety [84]. A review of management and treatment of victims of IPV included possible interventions for PTSD [85]. The importance of follow-up with patients who are experiencing family violence cannot be overemphasized.

### 5.3. Reporting Child Maltreatment

NICE guidance [86] acknowledges that different actions are required depending on whether the provider ‘considers’ maltreatment (maltreatment is one possible explanation for the sign or symptom) or ‘suspects’ maltreatment (the provider has a serious level of concern about maltreatment, but not necessarily proof of it). In considering maltreatment, the provider should look for other signs or symptoms of maltreatment in the child’s history, presentation or caregiver-child interactions now or in the past. Considering maltreatment may also involve following up with the child in a reasonable timeframe. Providers need to be aware of their own and the child/family’s culture when considering maltreatment, including any stereotypes, assumptions, or (lack of) cultural knowledge [62], while also recognizing that a cultural practice “should not justify hurting a child or young person” [28]. For example, when social norms that condone harsh parenting practices mirror provider or parental norms and values it can be hard to recognize and respond to maltreatment [87].

Suspecting maltreatment entails, where mandatory reporting is in place, that the provider reports their suspicion to child protection services, and where policies or procedures are in place, to refer the child to children’s social care. Providers need to be aware of the specific legislation in their area of practice. It is important to inform the caregiver about the need to report to child protection when considered safe to do so. Particularly in those circumstances when a child has disclosed exposure to IPV in the home, and a non-offending caregiver is accompanying the child, it can be helpful to involve the caregiver in a report to child protection services [51]. In contexts where child welfare or social service responses are not well established, it is important for providers to balance their ethical duty to report with the best interests of the child [88]. In contexts where it is not safe to report a child or where child welfare or social service responses are not available, clinicians may need to assess for the availability of support from someone trustworthy in the child’s life, such as a teacher, pediatrician, or grandparents. Consultation with experts in children’s health or with child welfare can assist clinicians in making decisions about the appropriateness of a referral to child welfare versus other preventive or supportive services. When referral is a legislative requirement, clinician advocacy on behalf of the family (e.g., stating to child welfare how the clinician is planning to help the family and what the family is currently doing well) can be important to help mitigate potential negative effects of the referral. Regardless of the legislative, policy, or ethical obligation [88] for referral to child welfare, it is important for mental health providers to work with children and families to help ensure they are accessing services and supports to meet their needs. Reporting may be a legislative requirement—one that may at times lead to moral injury in healthcare and social service providers—whereas appropriate support is an ethical or moral responsibility.

## 6. Assessment and Interventions for Those Who Are Committing Violence

Very little research exists regarding strategies to inquire about and safely respond to those who are using violence in their relationships [89]. It is important to assess for perpetration, victimization, or both. If you are concerned or need to inquire further about potential use of violence in the relationship, it is generally good to start with general questions about the relationship, such as “Tell me about how things are between you and your (ex) partner” or “What challenges are you experiencing in this relationship”? [90]. If the person discloses the use of violence in their relationship, some things you could say might include: “Telling me about this is important” or “It sounds like you see the problems that violence is causing for you and your family and the importance of stopping” [51]. In cases of violence perpetration, it is important to assess for homicidal risk, including (if relevant) frequency, intensity, current plan, degree of lethality, available means, as well any deterrents to violence and social supports. IPV between adults is not reportable to the police unless a practitioner is concerned about a serious imminent risk to the patient or someone else (see above for mandatory reporting in some jurisdictions when a child is exposed to IPV between caregivers) [85].

Providers can also assess the impact of the violence on the person using the violence and anyone experiencing the violence (e.g., the partner or ex-partner, children), the person’s readiness to change, and any co-morbid or contributing problems (e.g., substance use, addiction, mental health concerns, employment concerns) [51]. In situations where you are seeing both partners (e.g., separate parent intake assessments before children’s therapy), it is important to be extra cautious about confidentiality and safety issues. Information about one partner should not be shared with the other, such as the partner’s disclosures of abuse, treatment of injuries, or other safety concerns. It is also important not to make statements that minimize violence, such as “relationships are difficult and we all get aggressive towards each other sometimes” [51]. This can be difficult when providers are validating emotions (e.g., “I can see you are feeling frustrated”), as some people are not able to easily distinguish between internal emotions (e.g., anger) and external behaviours (e.g., yelling). Providers should be mindful of this when commenting upon the patient’s interpretations of their emotions, thoughts, and behaviours.

Following assessment, intervention may be warranted. This might include providing meaningful information about the harmful impact of IPV or referral to evidence-based interventions for support of co-morbid problems (e.g., substance use, mental health) [89]. Batterer intervention programs show inconclusive results at this point in time [91]. Caution is also required when considering couples’ interventions, which are generally not recommended due to a concern about increased risk of danger to the person experiencing abuse. Findings from a recent review emphasize that positive findings can only be applied to “mild to moderate situational couple violence” [92]. Assessing what counts as mild or moderate situational couple violence is difficult and often subjective; consultation with an IPV expert is recommended when developing competence around assessments of the usefulness of couples’ interventions for any particular couple. For example, it may be warranted to have a team discussion about the appropriateness of a couples’ intervention. When the clinician assesses each person separately, they may need to communicate with them about the impact of violence on mental health and also to indicate that violence is never justified and is never a valid relationship option. The person experiencing violence in the relationship may also need support understanding that violence is never justified, especially when there are societal, community, or cultural norms (e.g., patriarchy) that justify violence against women [93]. For example, women in many patriarchal cultures have internalized ideas about what is and is not violence and may believe that the violence they are experiencing is justified. Giving a clear message about the unacceptability of violence in any relationship helps in these situations. In addition to available services, there is a need to develop specialized services for specific populations who are at high risk for experiencing violence. For example, one meta-analysis indicated that for women with severe mental illness, prevalence rates of IPV in the past-year ranged from 15 to 22 percent [9,94].

## 7. Training Resources for Mental Health Providers

A number of resources have been developed to support providers with patients/clients who are experiencing family violence. It is important for providers to become familiar with these resources and resources in their community during training and not wait until a patient with experiences of family violence presents to their practice.

In order to assist healthcare and social service providers in recognizing and safely responding to family violence, the VEGA Project has created pan-Canadian guidance and educational resources for providers called VEGA Family Violence Education Resources [95]. These online educational resources were developed in consultation with 22 national organizations, with funding from the Public Health Agency of Canada. While derived for Canada, the evidence and responses for recognizing and safely responding have a broad applicability. The VEGA Family Violence Education Resources focus on child maltreatment and IPV and were developed based on extensive systematic reviews on these topics. The child maltreatment reviews were conducted in coordination with the WHO officials and in parallel to WHO child maltreatment guidance development processes [96]. VEGA includes a platform of evidence-based guidance and accredited curriculum comprised of learning modules (e.g., care pathways, scripts, how-to videos), interactive educational scenarios, and a handbook [95]. The project created a common approach to recognition and safe response to IPV and child maltreatment (see Table 1).

The World Psychiatric Association has developed an international competency-based curriculum for responding to IPV [97]. This curriculum was developed under the leadership of Dr. Donna Stewart and Dr. Prabha Chandra and with the support of a steering group of six experts from across the globe (three from WHO) and two educational consultants. This curriculum addresses competencies for medical students, psychiatry trainees, and psychiatrists regarding IPV and sexual violence, including definitions, prevalence, misconceptions, health sequelae, assessment, psychological first aid, resources, documentation, and psychiatric management of related mental health traumas. The WPA curriculum is comprised of several educational tools, including WHO clinical and policy guidelines, a WHO clinical handbook, key paper abstracts, a list of books, manuals and toolkits, a teaching set of slides on IPV and sexual violence, and international case vignettes with a quiz [98].

Other relevant training resources include the WHO clinical handbook for IPV [68]; the WHO health sector guidelines for responding to child maltreatment [96]; the Australian ‘White Book,’ which summarizes guidance for general practitioner responses to abuse and violence experience in patients [99], and the United Kingdom’s Violence Abuse and Mental Health Network [100].

While the above resources are useful, there are many potential areas for service improvement. Many mental health professionals are not aware of available resources for patient/clients experiencing IPV or maltreatment. Conversely, many specialized IPV and maltreatment services are not well trained in mental health services [101,102]. There is a greater need for collaboration and cross-disciplinary information sharing and training across sectors [102,103,104]. Mental health providers can also play important roles in “train the trainer” [105] strategies to facilitate the uptake of mental health knowledge and awareness of rights-based and trauma- and violence-informed care principles in other healthcare settings, such as emergency rooms, primary health care settings, and (lay) community health services.

## 8. Conclusions

Family violence, including child maltreatment and IPV, are complex exposures that require a safe, trauma- and violence-informed response from providers. A number of elements of recognition and safer responses are discussed in this article, including but not limited to signs, symptoms and risk factors of family violence; asking about potential family violence in a way that acknowledges the clinical presentation of the person (e.g., starting with the person’s presenting concern and general well-being, tailoring questions to the person’s presentation); and responding to family violence in a culturally informed way that conveys an understanding of the patient/client’s experience and offers them emotional and practical support. In addition, training resources for mental health providers are summarized and drawn upon throughout. While family violence is a complex and difficult issue, safer responses from mental health providers can be an important part of the patient’s experience of moving towards a life free from violence.

## Figures and Tables

**Table 1 ijerph-19-15672-t001:** VEGA principles for a common approach to recognizing and safely responding to family violence [51].

Care Pathway	Common Approach
Trauma- and violence-informed care	Trauma- and violence-informed care is a useful way to approach the provider-patient care encounter as it brings to awareness the high prevalence of violence experiences in patient’s lives and acknowledges the overlapping and additive effect of violence in people’s lives.
Recognizing family violence	Any time providers are concerned about a patient’s social, emotional, or behavioural responses, they should consider what environmental factors, including family violence, could be influencing these responses. Some patients show no signs or symptoms of family violence. For patients who do show signs and symptoms, they are usually not sufficient to confirm family violence. To move from considering to suspecting family violence, further inquiry about a sign or symptom is often required.
Asking about family violence	Before inquiring about potential family violence, certain conditions of safety must be achieved, such as a private space for discussion (this includes considerations of privacy for children). Safe inquiries about potential family violence experience must involve discussions about the limits of confidentiality and ideally follow a phased approach to questioning (first asking about the presenting problem, then about general well-being, then about safety in the home if needed).
Safe responses to disclosures of family violence	Safe responses to disclosures of family violence build upon the provider’s communication skills and at the very least convey an understanding of the patient’s experience and offers of support. In addition to immediate safety, it is important to consider an assessment of the patient’s physical and mental health symptoms, as well as any academic/workplace or social problems and to offer a referral to evidence-based services as needed. In jurisdictions where mandatory reporting is in place, if providers suspect child maltreatment (including children’s exposure to intimate partner violence), they are required to make a report to child protection services. Caregivers should be involved in the report when it is safe to do so.
Documentation	Providers should accurately and completely document findings of their interaction with patients for the purposes of follow-up and to support patients in accessing appropriate services.
Used with permission by the © 2020 VEGA Project, McMaster University-All rights reserved. For more detailed information, please see the VEGA Family Violence Education Resources.	

## Data Availability

Not applicable.

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
