# Peer review of "Child Maltreatment and Intimate Partner Violence in Mental Health Settings"

_ijerph, 2022, doi:10.3390/ijerph192315672_

Round 1
Reviewer 1 Report
Overall, this is a thorough review of the issues surrounding IPV and child maltreatment for various groups including women and children, and additional subgroups such as those identifying as LGBTQ+. The paper does good job of highlighting the struggles, describing different programs/approaches that exist to address issues with IPV. The manuscript also addresses the weaknesses of such approaches, and makes various evidence based suggestions of what can be done to be more effective at addressing issues related to IPV and child maltreatment.
The paper itself Is a strong review of existing research with well supported arguments for what else can be developed to address these major issues. It is a strong review paper, and would be of interest to many readers.
Author Response
We are thankful for the Reviewers’ comments. We note that this journal posts responses to Reviewers’ separately. Below is Reviewer 1’s comments in bold and our responses in bullet points underneath.
Overall, this is a thorough review of the issues surrounding IPV and child maltreatment for various groups including women and children, and additional subgroups such as those identifying as LGBTQ+. The paper does good job of highlighting the struggles, describing different programs/approaches that exist to address issues with IPV. The manuscript also addresses the weaknesses of such approaches, and makes various evidence based suggestions of what can be done to be more effective at addressing issues related to IPV and child maltreatment.
The paper itself Is a strong review of existing research with well supported arguments for what else can be developed to address these major issues. It is a strong review paper, and would be of interest to many readers.
- Thank you for your positive comments regarding our commentary. We also hope that it will be of interest to many readers.
Reviewer 2 Report
Dear Authors,
Thank you for the opportunity to review this exciting paper entitled " Child Maltreatment and Intimate Partner Violence in Mental Health Settings." It is an important subject, and good to do it. The literature review was well-written and laid out the issues in the paper well. Some points need to be revised, and my suggestions are as follows:
1. This manuscript mainly talks about “Child Maltreatment and Intimate Partner Violence (IPV)” in family violence, but IPV includes violence in unmarried intimate relationships, such as dating violence; the types of marital violence and dating violence are not the same (the latter consists of online dating violence), and the situation of the two types of victims are different, the treatment will be other. Hence, it is recommended to change the title of “Intimate Partner Violence” to “Domestic Violence or Marital Violence” or point out in the text that the IPV discussed in this manuscript is limited to family violence.
2. The second paragraph of the second page talks about gender-based violence. It is recommended to add dating violence to explain the types of IPV. In addition to marital violence, it includes online and offline dating violence. Briefly describe them. Suppose the authors do not want to change the title “Intimate Partner Violence.” In that case, the authors can use the last sentence here to explain that the IPV in this manuscript mainly discusses marital violence, which makes the following descriptions and discussions more in line with the author’s subject scope (family violence).
Author Response
We are thankful for the Reviewers’ comments which we believe have improved the manuscript. We note that this journal posts responses to Reviewers’ separately. Below is Reviewer 2’s comments in bold and our responses in bullet points underneath.
Dear Authors,
Thank you for the opportunity to review this exciting paper entitled " Child Maltreatment and Intimate Partner Violence in Mental Health Settings." It is an important subject, and good to do it. The literature review was well-written and laid out the issues in the paper well. Some points need to be revised, and my suggestions are as follows:
- Thank you for your positive comments about our commentary.
- This manuscript mainly talks about “Child Maltreatment and Intimate Partner Violence (IPV)” in family violence, but IPV includes violence in unmarried intimate relationships, such as dating violence; the types of marital violence and dating violence are not the same (the latter consists of online dating violence), and the situation of the two types of victims are different, the treatment will be other. Hence, it is recommended to change the title of “Intimate Partner Violence” to “Domestic Violence or Marital Violence” or point out in the text that the IPV discussed in this manuscript is limited to family violence.
- We appreciate the Reviewer’s comments about the differences between, for example, online dating violence and violence in a domestic/intimate relationship. We use “intimate partner violence” to refer to violence across intimate relationships because a) violence in these contexts often happens outside of the bounds of marriage (making “marital violence” too narrow a term) and b) violence in these relationships may also occur outside of a domestic setting (for example, intimate partner violence often continues on and in many cases can become more serious/dangerous after partners are no longer living together (making “domestic violence” too narrow a term). However, we appreciate that our commentary does not address dating violence between teenagers, nor does it address online/cyber violence, both of which have their own important considerations. And while some of the educational resources we discuss mention increasing technological violence between partners and ex-partners (e.g., a partner using ‘smart technology’ in the home to terrorize their partner), this is not the primary focus of the educational resources, nor is it the focus of this manuscript. Indeed, increasing research and support in these areas (e.g., online dating violence) is needed. In addressing the Reviewer’s suggestion, we have tried to clarify our definition of intimate partner violence in the first paragraph, as well as to indicate some important types of violence that we do not address in the commentary (such as teen dating violence, cyber violence, elder abuse, and child sex exploitation).
- The second paragraph of the second page talks about gender-based violence. It is recommended to add dating violence to explain the types of IPV. In addition to marital violence, it includes online and offline dating violence. Briefly describe them. Suppose the authors do not want to change the title “Intimate Partner Violence.” In that case, the authors can use the last sentence here to explain that the IPV in this manuscript mainly discusses marital violence, which makes the following descriptions and discussions more in line with the author’s subject scope (family violence).
- We agree that dating violence and online dating violence are also gendered. Since we have added a sentence noting that we do not address dating violence/online violence just before this paragraph, we have not explained dating violence/online violence in this paragraph, so as to not confuse readers. We hope this explanation is acceptable.
Reviewer 3 Report
I sincerely congratulate the authors on this excellent manuscript. This is the first time I directly accept a manuscript without any comment. I feel that more manuscripts like this one are needed. The authors have combined scientific preciseness with specific advice on dealing with people who present IPV or child maltreatment. I'm sure this paper will be a reference for scientists and practitioners.
Author Response
We are thankful for the Reviewers’ comments. We note that this journal posts responses to Reviewers’ separately. Below is Reviewer 3’s comments in bold and our responses in bullet points underneath.
I sincerely congratulate the authors on this excellent manuscript. This is the first time I directly accept a manuscript without any comment. I feel that more manuscripts like this one are needed. The authors have combined scientific preciseness with specific advice on dealing with people who present IPV or child maltreatment. I'm sure this paper will be a reference for scientists and practitioners.
- We express our sincere thanks for your positive feedback on our commentary.
Reviewer 4 Report
The article gives an overview of existing knowledge concerning best practices in recognizing and responding to (suspicions of) child maltreatment and interparental violence. The authors have done a great job on this commentary. The commentary is especially relevant for people in the clinical field as it offers clear and practical information about best-practice elements. The manuscript is well-written and includes a reasonably complete and up-to-date literature review. I don't have many comments for the authors, but solely have a few remarks about the Evidence-based interventions paragraph. The part that focuses on children is quite generic, though children's age and developmental stage (eg infants/toddlers versus adolescents) ask for different approaches and interventions. For example, the authors' statement that "If caregivers are not able to be an emotionally safe and regulated person in the room with the child, child-only therapy may be indicated". This might be true for older children, but especially for very young children the presence of a stable and emotionally safe caregiver is essential. In addition, parent-child interventions such as attachment-based interventions are not mentioned in this paragraph even though effectiveness has been clearly demonstrated for maltreatment populations (such as the AVI, see Moss et al and Cyr et al). I can understand that it was not the goal of this commentary to provide a comprehensive review about evidence-based interventions, but think that at least the attachment perspective and relational approach should be mentioned here.
Author Response
We are thankful for the Reviewers’ comments which we believe have improved the manuscript. We note that this journal posts responses to Reviewers’ separately. Below is Reviewer 4’s comments in bold and our responses in bullet points underneath.
The article gives an overview of existing knowledge concerning best practices in recognizing and responding to (suspicions of) child maltreatment and interparental violence. The authors have done a great job on this commentary. The commentary is especially relevant for people in the clinical field as it offers clear and practical information about best-practice elements. The manuscript is well-written and includes a reasonably complete and up-to-date literature review.
- Thank you for your positive comments about our commentary.
I don't have many comments for the authors, but solely have a few remarks about the Evidence-based interventions paragraph. The part that focuses on children is quite generic, though children's age and developmental stage (eg infants/toddlers versus adolescents) ask for different approaches and interventions. For example, the authors' statement that "If caregivers are not able to be an emotionally safe and regulated person in the room with the child, child-only therapy may be indicated". This might be true for older children, but especially for very young children the presence of a stable and emotionally safe caregiver is essential. In addition, parent-child interventions such as attachment-based interventions are not mentioned in this paragraph even though effectiveness has been clearly demonstrated for maltreatment populations (such as the AVI, see Moss et al and Cyr et al). I can understand that it was not the goal of this commentary to provide a comprehensive review about evidence-based interventions, but think that at least the attachment perspective and relational approach should be mentioned here.
- The Reviewer has raised a very important point about the limitations of this section. We hoped to summarize some of the important principles of selecting an intervention based on the results of an assessment, but did not intend to provide an exhaustive summary of evidence-based interventions. We had the privilege of reviewing Moss et al.’s RCT for our guidance development (for VEGA and in our parallel work with WHO) and listed it as promising but could not recommend it at the time. For the GRADE process for guidance development we often needed more than 1 positive RCT to recommend. The conclusion, based on VEGA’s evidence review process, which used GRADE, that we could not recommend an attachment intervention was a big struggle for researchers/clinicians during guidance development as we agree with the Reviewer that, especially for very young children, the therapeutic goal is often to increase the safe attachment behaviours of the caregiver. In addition, much important research has come out since our guidance development processes, such as the important work by Cyr et al. (although, again, given the small sample size we probably still could not recommend this intervention based on use of the GRADE procedures). In addition, as noted by the Reviewer, a recent systematic review by O’Hara has pooled a number of attachment interventions (including Moss) and was able to find positive results for parental sensitivity. We have cited this review and have added a few clarifying statements in this paragraph to speak to these important points raised by the Reviewer.